# Genetic Elements Orchestrating *Lactobacillus crispatus* Glycogen Metabolism in the Vagina

**DOI:** 10.3390/ijms23105590

**Published:** 2022-05-17

**Authors:** Rosanne Hertzberger, Ali May, Gertjan Kramer, Isabelle van Vondelen, Douwe Molenaar, Remco Kort

**Affiliations:** 1Department of Molecular Cell Biology, Vrije Universiteit Amsterdam, De Boelelaan 1108, 1081 HZ Amsterdam, The Netherlands; r.y.hertzberger@vu.nl (R.H.); i.r.van.vondelen@student.vu.nl (I.v.V.); d.molenaar@vu.nl (D.M.); 2ARTIS-Micropia, Plantage Kerklaan 38-40, 1018 CZ Amsterdam, The Netherlands; ali.may@dsm.com; 3Biodata & Translation Innovation, Royal DSM, Alexander Fleminglaan 1, 2613 AX Delft, The Netherlands; 4Laboratory for Mass Spectrometry of Biomolecules, University of Amsterdam, Science Park 904, 1098 XH Amsterdam, The Netherlands; g.kramer@uva.nl

**Keywords:** *Lactobacillus crispatus*, amylopullulanase, glycogen

## Abstract

Glycogen in the female lower reproductive tract is a major carbon source for colonization and acidification by common vaginal *Lactobacillus* species, such as *Lactobacillus crispatus*. Previously, we identified the amylopullulanase encoding gene *pulA* of *Lactobacillus crispatus* to correlate with the ability to autonomously utilize glycogen for growth. Here, we further characterize genetic variation and differential regulation of *pulA* affecting the presence of its gene product on the outer surface layer. We show that alpha-glucan degrading activity dissipates when *Lactobacillus crispatus* is grown on glucose, maltose and maltotriose, in agreement with carbon catabolite repression elements flanking the *pulA* gene. Proteome analysis of the S-layer confirmed that the amylopullulanase protein is highly abundant in an S-layer enriched fraction, but not in a strain with a defective amylopullulanase variant or in an amylopullulanase-sufficient strain grown on glucose. In addition, we provide evidence that *Lactobacillus crispatus pulA* mutants are relevant in vivo, as they are commonly observed in metagenome datasets of human vaginal microbial communities. Analysis of the largest publicly available dataset of 1507 human vaginal metagenomes indicates that among the 270 samples that contain a *Lactobacillus crispatus*
*pulA* gene, 62 samples (23%) had a defective variant of this gene. Taken together, these results demonstrate that both environmental, as well as genetic factors explain the variation of *Lactobacillus crispatus* alpha-glucosidases in the vaginal environment.

## 1. Introduction

The vaginal environment of reproductive-age women is unique amongst vertebrates in several aspects: it has a low pH due to high levels of lactate (~100–150 mM) [1], it has abundant glycogen levels [2], and its microbial community is dominated by *Lactobacillus* species [3,4]. In recent years, more evidence has emerged that the vaginal microbiome is associated with reproductive and sexual health. Women with a *Lactobacillus crispatus* microbiome are at lower risk of vaginal infections and sexually transmitted diseases [5,6,7]. In addition, vaginal *L. crispatus* is inversely associated with more serious health outcomes, such as subfertility [8,9], preterm birth [10] and persistent HPV infection [11,12].

We previously demonstrated that most vaginal *L.*
*crispatus* isolates are capable of autonomous growth on glycogen. A disrupted N-terminal signal peptide sequence in the amylopullulanase encoding *pulA* gene, either by small mutations or structural variants, coincided with the inability of growing on glycogen [13].

A recent key study [14] confirmed alpha-glucosidase activity of glycogen-degrading enzymes from different vaginal bacteria by heterologous expression. It showed that the *L. crispatus* amylopullulanase degrades glycogen, pullulan and amylose retaining activity at pH values that are common in a *Lactobacillus*-dominated vaginal environment. *L. crispatus* strains isolated from low-glycogen environments, such as the human or poultry gut, more often lacked the *pulA* gene indicating niche-specificity [15].

An open question in the field is the role of vaginal glycogen stores in bacterial acidification and the origin of vaginal glycogen degrading enzymes. Vaginal samples show alpha-glucosidase activity, which was postulated to be mostly of human origin on the basis of antibody [16,17] and proteomic analysis [18]. At the same time, results from two studies have indicated that alpha-glucosidases in a subset of vaginal samples may be of bacterial origin, on the basis of optimal pH and substrate and product profiling [14,19]. However, a metaproteomic study of vaginal lavages from women with a *L. crispatus*-dominated microbiome failed to detect any *L. crispatus* amylopullulanase protein, which poses questions about host-related or genetic factors influencing its presence [18].

Here we present results showing suppression of PulA activity in glucose-, maltose- and maltotriose-grown cultures by enzymatic assays for alpha-glucosidase. We use a mass spectrometric approach to demonstrate the presence of the *L. crispatus* amylopullulanase enzyme in an S-layer enriched fraction and its loss in the presence of glucose or in a strain with a defective *pulA* gene. In addition, the analysis of a large vaginal metagenomic dataset demonstrates the presence of a significant number of mutations and structural variants, which are expected to fully disrupt amylopullulanase functionality in vivo, indicating that both environmental, as well as genetic factors, may restrainvaginal *L. crispatus* from metabolizing glycogen.

## 2. Results

### 2.1. Substrate Utilization of PulA-Sufficient and Deficient Lactobacillus crispatus Strains

Previously, we reported a collection of *L. crispatus* strains with variable growth on glycogen associated with the presence of an intact amino-terminal signal peptide for protein secretion in the *pulA* gene product (PulA) [13]. Here we further explore the substrate range of a *L. crispatus* deficient (RL09, hereafter referred to as *pulA*^−^) and sufficient (RL10, hereafter referred to as *pulA*^+^) strain.

To verify the identified deletion of two nucleotides, we used a PCR-amplified 300 bp DNA fragment encoding the *pulA* N-terminal region (including 42 nucleotides upstream the start site) of strains RL09 and RL10 and resequenced with Sanger dideoxy sequencing. This confirmed the presence of the deletion of two nucleotides in the signal peptide sequence of RL09 compared to RL10 (Appendix B Figure A1).

Previously, we found the *pulA*^−^ strain RL09 to be unable to grow on glycogen. We reassessed the growth and acidification of this strain and the *pulA*^+^ strain RL10 on NYCIII medium without glucose supplemented with 11 different carbohydrates to analyze other differences between the strains in their metabolic profiles. Optical density and medium pH after 24 h of anaerobic growth are shown in Figure 1A,B, respectively.

Both *L. crispatus* strains were equally capable of growth on all carbon sources, except for lactose, raffinose, cellobiose and, in disagreement with previous findings, lactulose [20]. Growth was accompanied by acidification of the medium to a pH of 3.9, corresponding to the in vivo vaginal pH [1] and the pKa of lactic acid. Glycogen, both of oyster and bovine origin, was the only carbohydrate resulting in a clear difference between the strains: the *pulA*^−^ deficient strain grew to an optical density of ~0.5 and acidified to a pH of 6.0 (Oyster glycogen) and 5.5 (Bovine glycogen) possibly due to the presence of smaller maltodextrins and glucose after autoclaving the glycogen. The growth of the *pulA* sufficient strain on glycogen was comparable to that on glucose, acidifying the medium to a pH of 3.9. Regarding growth on the 11 carbohydrates tested, we conclude that *pulA*^+^ and *pulA*^−^ strains do not substantially differ in their metabolic capacities apart from the ability to grow on glycogen.

### 2.2. Starch Degradation Activity of PulA Sufficient and Deficient L. crispatus Strains on Various Carbon Sources

Next, we measured the enzymatic activity of the amylopullulanase enzyme to understand its regulation and to find amylopullulanase-expressing conditions that would allow for a direct proteomic comparison between the two strains. Genetic elements in the locus of the *pulA* gene are strongly indicative of carbon catabolite repression, similar to the regulation of amylopullulanases in other Lactobacilli [21]. The promoter region of the gene has a *cre* (catabolite responsive element) like palindromic sequence (TGTTATCGATAACA). The catabolite responsive element contains a well-known binding site for the global regulator CcpA (carbon catabolite protein A), which is known for suppressing metabolic pathways for alternative carbohydrates in the presence of glucose in various *Lactobacillus* species [22,23,24]. Downstream of the *pulA* gene, we identified two open reading frames with homology to the LacI family of repressors, containing a DNA binding domain in one open reading frame (1–110) and an effector domain (91–219) in the other, possibly involved in catabolite repression through binding of the breakdown products of glycogen (Figure 2).

To verify the predicted regulation by carbon sources, we grew the *pulA* sufficient strain *L. crispatus* to stationary phase on NYCIII medium supplemented with glycogen, maltotriose, maltose, glucose, or galactose, and measured alpha-glucosidase activity in the pellet and supernatant. We included galactose since (i) lactic acid bacteria generally show a preference for glucose over galactose [25,26], and (ii) we hypothesize that galactose does not repress *pulA* expression to the same extent as glucose.

After 48 h of growth, spent medium and bacterial cells were mixed with a starch solution to detect starch degradation in a standard starch-iodine assay. Starch was used as a substitute for glycogen since it can be easily detected using iodine and has an amylopectin moiety with alpha1–4 and alpha1–6 linkages comparable to glycogen. Total starch degradation for a period of 24 h at 37 °C was used as a measure of alpha-glucosidase activity in the sample. Starch degradation was observed in the pellet (Figure 3) as well as the spent growth media (Appendix B Figure A2) of *L. crispatus pulA*^+^ strain RL10 after growth on glycogen and galactose, but not after growth on glucose, maltose and maltotriose, confirming the repression of this gene under these conditions.

Alpha-glucosidase activity in cells and spent medium of the *pulA*^+^ strain was comparable to that of the *pulA*^−^ strain on all substrates except for glycogen since the *pulA*^−^ strain did not sufficiently grow on glycogen to measure starch degradation. Bacterial cells or supernatants of the *pulA*^−^ strain grown on glucose, maltose, maltotriose, or galactose did not degrade any starch confirming the link between the defective amylopullulanase enzyme and glycogen metabolism.

### 2.3. S-Layer Enrichment and Proteome Analysis

In our previous study, we identified a surface layer associated protein (SLAP) domain in the *L. crispatus* amylopullulanase protein sequence [13], which prompted us to investigate its localization in the surface layer. We hypothesized that the natural variation in the amino-terminal signal peptide for secretion of amylopullulanase would result in the presence or absence of an active amylopullulanase in the S-layer. Furthermore, we wanted to confirm the catabolite repression of amylopullulanase by glucose, as observed in this study by use of the starch degradation assay (Figure 3), would be reflected by the levels of amylopullulanase in the S-layer. We utilized galactose as a carbon source since it (i) allows for the growth of both *pulA*^−^ and *pulA*^+^ strains and (ii) is a condition under which we observed cell-associated starch degradation activity (Figure 3). We compared cells of the *pulA*^+^ RL10 strain grown on galactose, glucose, and glycogen, in order to verify the expression of amylopullulanase in an S-layer enriched fraction under these conditions.

The S-layers form the outermost structure of the cell envelope and represent up to 15% of the total protein content [27]. As the S-layer protein subunits (SLPs) and S-layer associated proteins (SLAPs) are non-covalently linked to the cell wall, we released the S-layer fraction from the cells and let it disintegrate into monomeric polypeptides by exposure to high concentrations of the denaturing agent LiCl. Mass-spectrometric analysis of this isolated S-layer fraction led to the identification of peptides of at least 28 SLAP-domain containing proteins, including a number of abundant SlpA-domain containing proteins (Table 1). The most prominent distinction in the amount of all proteins identified in the RL09 and RL10 S-layers is a 90-fold difference for amylopullulanase The minor background of amylopullulanase peptides in RL09 may result from lysed cells releasing intracellular amylopullulanase. S-layer fractions of the *pulA*^+^ strain RL10 grown on glucose showed a 29-fold lower amylopullulanase presence compared to fractions from glycogen grown cells. Two uncharacterized surface layer proteins (A0A7T1TNA6 and A0A125P6L8) are strongly upregulated in the presence of glucose by a factor of over 200; all other proteins show minor differences in levels with ^2^log ratios between ranging from −2.2 to 1.4 The overall fraction of S-layer proteins appear to have been reduced in the presence of glucose (0.2 versus 0.4–0.5 in the other three fractions).

### 2.4. Metagenome Analysis

Our previous comparative genomics analysis of a collection of *L. crispatus* strains isolated from the outpatient clinic in Amsterdam revealed variations in the N-terminal signal peptide region of the *pulA* gene that correlated with the ability of a strain to grow on glycogen [13]. In this strain collection, we found seven amylopullulanase-deficient strains sharing four different types of disruptions of the N-terminal sequence (Table 2). Although not all published *L. crispatus* genomes show the presence of a *pulA* gene [15], the N-terminal disruptions of the *pulA* sequence observed in our collection were not found in other *L. crispatus* genomes reported thus far.

We hypothesize that the isolation of these *pulA* variants is not a result of selective enrichment during the isolation procedure, but that *L. crispatus* strains with defective *pulA* genes are common in the vaginal microbiota. To test this hypothesis, we assessed the prevalence and frequency of disruptive *pulA* gene sequence variants, such as those observed in our strain collection (Table 2) in a vaginal metagenome dataset of 1507 samples [28]. To our knowledge, this is the largest publicly available vaginal microbiome data collection, which includes samples from several different studies of North American, African, and Chinese women. In total, we analyzed 28.6 billion sequencing reads that made up 3.9 tera base pairs (Tbp) of metagenome sequencing data (on average, 19 million reads and 2.6 Gbp data per sample).

We first determined the prevalence of small but disruptive variations across the *pulA* gene sequence, similar to those observed in *pulA*^−^ strains RL02 and RL09 (Table 2). Out of 1507 samples we aligned to the *L. crispatus* RL10 *pulA* gene sequence, 270 resulted in alignments with a minimum of 10 times coverage. Variant calling on these 270 samples resulted in a total of 3202 unique variants. After low-frequency (allele frequency lower than 0.05) and low prevalence (observed in less than three samples) variants were filtered out, 54 variants remained, of which 19, 28 and 6 were low (e.g., synonymous mutations), moderate (e.g., mutations that cause an amino acid change), and high impact (stop gain or frameshift mutations), respectively (Figure 4).

In 15 out of 270 samples (5.5%), at least one of the high-impact mutations was observed at a high allele frequency (>0.7), where we expect an amylopullulanase deficient phenotype (Appendix A). Three of these high-impact mutations were near the N-terminus of the *pulA* sequence at positions 156, 207, and 225, impacting 11 samples in total.

Subsequently, we analyzed the same set of 270 samples for larger structural variants compared to the pulA^+^ RL10 *pulA* gene sequence. In 56 out of 270 samples (20%), we identified large structural variants which had a high frequency (>0.7) (Appendix A). The disruptive structural variation in 17 out of 56 of these samples was found to be due to the insertion of transposase sequences into the amylopullulanase gene sequence, as in the case of the *pulA* locus from strains RL06, RL07, RL19, and RL26 in our own strain collection. Taken together, we identified high-frequency mutations in 62 of 270 *L. crispatus pulA* samples, which is indicative of a dysfunctional amylopullulanase of the dominant strain and the inability of *L. crispatus* to contribute to the acidification of the vaginal environment with glycogen as a carbon and energy source.

## 3. Discussion

In this study, we focus on genetic variation and disruption of the *Lactobacillus crispatus pulA* gene. Amylopullulanase is thought to play a central role in vaginal metabolomics. It catalyzes the first step of a metabolic pathway turning the most abundant vaginal carbohydrate (glycogen) into the most abundant metabolite (lactate) and is produced by one of the most abundant vaginal species *L. crispatus*. In a previous study, we published 33 genomes of vaginal *L. crispatus* isolates that displayed remarkable variation in the gene locus. Strains with a disrupted *pulA* gene, through small indels or structural variants, were incapable of growth on glycogen.

The wealth of publicly available metagenome datasets, repositories, and gene catalogs allows for a more hypothesis-driven research approach concentrating on specific genes that are associated with in vivo functionality. In the largest available metagenome database, a substantial number of samples had similar genetic variations compared with our previously reported whole-genome sequences of *L. crispatus* isolates. These variations consisted of both small deletions as well as relatively large structural variations disrupting the functionality of this gene.

Disruptions in the secretome of catalytically active enzymes are a well-known phenomenon in the microbial world. Bacteria may take advantage of cells in close proximity that cleave ’public goods’, such as proteins or glycans into smaller peptides or maltodextrins. For instance, a protease-positive *Lactococcus lactis* population gradually lost protease activity in the entire population in a spatial-dependent manner. In this case, the ‘cheating’ proteolytic negative strains can utilize peptides supplied by the proteolytic positive strains without having the cost of protease production [29]. These disruptions may provide a fitness advantage to isolates, especially when, as we show in our mass spectrometric analysis, the enzyme forms a substantial fraction of the outer surface layer.

In the vaginal metagenome dataset we analyzed here, we did not find evidence of an extensive variety of amylopullulanase sequences within a single vaginal microbiome: most samples only showed the presence of one variant, which we assume coincides with the dominant presence of one strain. We hypothesize that the metabolic advantage that a cheating strain in complex ecological samples may experience, is not so much a matter of intraspecies competition, but rather due to interspecies cross-feeding of metabolites. The presence of other glycogen degrading bacteria or human amylase may create an environment in which a *L. crispatus* strain has energetic advantages from shutting down the expression and secretion of S-layer associated enzymes while benefiting from the breakdown products of extracellular glycogen hydrolysis by other species.

A recent study did not find such evolutionary pressure to mutate the amylopullulanase: even after cultivation of up to 1000 generations by sequential sub-culturing in synthetic vaginal fluid, no genetic variations in the *pulA* locus of strains of *L. crispatus* were detected [30]. We hypothesize that this may be due to the available carbohydrates in this simulated vaginal fluid, which contains both glucose and glycogen and is further supplemented with amylase which degrades the glycogen into smaller maltodextrins. The results of our alpha-glucosidase assays and proteome analysis imply that *L. crispatus* grown in such an environment will first grow on glucose, maltose and maltotriose and will suppress amylopullulanase expression until these carbohydrates are depleted. This will diminish the advantage of cheating or any other evolutionary pressure on the production of alpha-glucosidase.

The genetic disruption affected a minority of vaginal samples in the metagenomic dataset we analyzed (23%) and a minority of isolates in our strain collection. In most environments, *L. crispatus* maintains an intact amylopullulanase gene, even though disruption of the *pulA* gene occurs at relatively high frequencies. A recent metagenomics study and targeted PCR by Lithgow et al. revealed that 36% of *L. crispatus*-dominated metagenomes from their African cohort lacked a functional *L. crispatus pulA* gene, and reported a three-fold higher frequency of gene loss than that seen in metagenomes from European and North American women [31]. However, this is not reflected by the prevalence of disrupted *pulA* genes among the known ethnic groups in the metagenome data set of 1507 vaginal samples we have analyzed (Appendix B Table A1). Our study shows that from 781 black or African American women, 102 contained a *pulA* gene, including 12 disrupted genes (12%), while from 163 white or Caucasian women, 55 contained a *pulA* gene, including 19 disrupted genes (34%).

Our study suggests that carbon catabolite repression may further reduce the activity of *L. crispatus* amylopullulanase activity in vivo. Bacteria generally regulate metabolic pathways, optimizing their growth strategy by prioritizing carbon sources based on nutritional quality. The presence of glucose, maltose and maltotriose in the vagina has been well-documented, especially in *Lactobacillus*-dominated environments [32,33].

However, given the positive relationship between amylopullulanase activity and smaller maltodextrins in the study by Lithgow et al. [31], we think further studies should include analyses of metatranscriptomes and carbohydrate concentration, to clarify whether the observed in vitro *pulA* regulation, depending on carbohydrate availability in *L. crispatus,* corresponds to its in vivo regulation. Based on the prevalence of these carbon sources in vaginal metabolomics studies, as well as human amylase, we expect that *L. crispatus* in the vaginal environment preferentially relies on glycogen cleavage by other bacteria or host amylase and will only express amylopullulanase when other carbon sources are depleted.

## 4. Materials and Methods

### 4.1. Standard Cultivation Conditions

*Lactobacillus crispatus* amylopullulanase deficient strain (*L. crispatus* RL09) and an amylopullulanase sufficient strain (*L. crispatus* RL10) were routinely cultivated on NYCIII medium with HEPES (2.4 g/L), proteose peptone #3 (Becton Dickinson Franklin Lakes, NJ, USA, 211693) (15 g/L), yeast extract (ThermoFisher Scientific Waltham, MA USA, LP0021) (3.8 g/L), NaCl (5 g/L), glucose monohydrate (Santa Cruz Biotechnology, Dallas, TX, USA 14431-43-7) (5.5 g/L g), heat inactivated horse serum (10%) and for agar plates 1,5% (*w/v*) agarose. Plates, precultures and subcultures were grown under anaerobic conditions (gas mixture of 10% CO_2_ and 90% N_2_ at 37 °C.

### 4.2. PCR and Sequencing of Signal Peptide Region L. crispatus Amylopullulanase

Previously, Illumina-whole genome sequencing showed a two nucleotide deletion in the N-terminal *pulA* signal peptide sequence of *L. crispatus* RL09. In order to confirm this frameshift mutation, we performed a genetic analysis of the area with deletion by PCR amplification and Sanger sequencing. Genomic DNA was isolated from a 24 h liquid culture using the GenElute^TM^ Bacterial Genomic DNA Kit (Merck, Darmstadt, Germany). A 300 base pair sequence at the start site of the amylopullulanase was amplified with a forward primer GCAAATGAAAGCGCATACGTTT (annealing 42 base pairs upstream the start site) and a reverse primer TGTTGACGCTGCTTTGCTT (annealing 257 base pairs downstream the start site) using a High Fidelity Phusion^®^ DNA polymerase (ThermoFisher, Waltham, MA, USA). The size of the PCR product was verified with ethidium bromide on a gel to be the predicted size of 299 bp. The PCR product was purified using the GeneJET PCR Purification Kit (ThermoFisher, Waltham, MA, USA) and sequenced by Sanger Dideoxy sequencing (Eurofins, Luxembourg City, Luxembourg).

### 4.3. Serial Propagation on Various Carbohydrates

Growth on various carbohydrate sources was tested with NYCIII without glucose supplemented with D-lactose monohydrate (Sigma-Aldrich, St. Louis, MO, USA), D-raffinose pentahydrate (Alfa Aesar Haverhill, MA, USA), D-cellobiose (Sigma-Aldrich, St. Louis, MO, USA), D-melibiose (Sigma-Aldrich, St. Louis, MO, USA), D-maltose monohydrate (Honeywell, Charlotte, NC, USA), D-trehalose dihydrate (Sigma-Aldrich, St. Louis, MO, USA), D-galactose (Sigma-Aldrich, St. Louis, MO, USA), lactulose (Sigma-Aldrich, St. Louis, MO, USA), glucose monohydrate (Merck, Darmstadt, Germany), bovine glycogen (Sigma-Aldrich, St. Louis, MO, USA) and oyster glycogen (Alfa Aesar Haverhill, MA, USA). The growth of *L. crispatus* RL09 and *L. crispatus* RL10 on these carbohydrate sources was examined by serial propagation on a glucose-free NYCIII medium supplemented with 5% (*w*/*v*) of the different carbohydrates. The serial propagation was executed using two passages of twentyfold sub-culturing in a 96-well plate. Between each passage, the cells were anaerobically grown for 24h in jars (ThermoFisher, Waltham, MA, USA). To monitor growth, optical density was measured at 600 nm in a sterile 96-well flat bottom plate in a spectrophotometer (Spectramax Plus 384; Molecular Devices, San Jose, CA, USA) after ten times dilution in sterile phosphate-buffered saline solution. Acidification of the medium by *L. crispatus* was assessed by pH measurements (HI 2210 pH meter, Hanna Instruments, Woonsocket, RI, USA) after every passage after 24 h of anaerobic growth. Statistical testing of optical densities and pH values was conducted with t-tests: two samples assuming unequal variations (α = 0.05, two-sided).

### 4.4. The Starch Degradation Assay

Glycogen is a polymer of α−1,4-linked and α−1,6-linked glucose residues. Starch degradation, measured in a basic starch-iodine assay, can be used as a proxy since the amylopectin moiety of starch has a similar branched structure with α−1,4-linked and α−1,6-linked glucose residues. To test α-glucosidase activity strains were grown for 24 h. Cells were spun down and resuspended in amylase buffer (100 mM Na-Acetate with 5 mM CaCl_2_, pH 5,3). 50 uL supernatant or resuspended pellet was added to 150 uL of 10 g/L starch solution (Sigma-Aldrich, St. Louis, MO, USA, S9765) in amylase buffer. Chloramphenicol (Sigma-Aldrich, St. Louis, MO, USA, C0378-25G) from a stock solution in ethanol was added to a final concentration of 10 ng/mL to prevent growth; 50 µL of the resuspended pellet of culture spent medium was mixed with 150 µL of this starch working solution and incubated at 37 °C for 24 h. To visualize residual starch, 10 uL was added to 290 uL of an iodine stock solution consisting of 0.2 g I2 and 2 g Kl in 100 mL 50 mM HCl. Absorption was measured at 600 nm in a 96-well plate reader. Experiments were carried out with a minimum of (two) biological replicates for RL09 on maltose and maltotriose; all other conditions had between three and nine replicates carried out on separate days in a total of four independent experiments.

### 4.5. Enrichment for Surface Layer (Associated) Proteins

The enrichment for SLPs and SLAPs was modified from a standard LiCl S-layer extraction protocol for *L. acidophilus* (Goh et al., 2009; Lortal et al., 1992). Bacterial cells of *L. crispatus* were grown to stationary phase (24 h), centrifuged at 2236 g for 10 min (4 °C), and washed twice with 25 mL cold PBS (Gibco), pH 7.4. Cells were agitated for 15 min at 4 °C following the addition of 5 M LiCl (ThermoFisher, Waltham, MA, USA). Supernatants, containing SLPs and SLAPs, were harvested via centrifugation at 8994 g for 10 min (4 °C) and transferred to a 6000–8000 kDa Spectra/Por molecular porous membrane (Spectrum Laboratories Rancho Dominguez, CA, USA) and dialyzed against cold distilled water for 24 h, changing the water every 2 h for the first 8 h. The dialyzed precipitate was harvested via centrifugation at 20,000× *g* for 30 min (4 °C).

### 4.6. Sample Preparation for Proteomics Analysis

Protein pellets were resuspended in 200 µL of digestion buffer (5 mM DTT, 50 mM NH_4_HCO_3_, pH 8.0) and incubated at 55 °C for one hour to reduce disulfide bridges. Then samples were treated with 15 mM iodoacetamide (IAA) to alkylate free cysteines in the dark for 45 min. Subsequently, samples were digested with trypsin (1:100 *w*/*w* protease:protein ratio) at 37 °C for 18 h. The mixture of peptides was desalted using C18 solid phase extraction columns (Agilent Technologies, Santa Clara, CA, USA) following the manufacturer’s protocol. The final peptide mixture was dried in a vacuum centrifuge and suspended in 0.1% formic acid in water (ULC-MS grade, Biosolve, Valkenswaard, The Netherlands) for MS analysis.

### 4.7. LC-MS/MS Analysis

Mass spectrometric analysis of 200 ng peptides was carried out on a timsTOF Pro (Bruker, Bremen, Germany) equipped with an Ultimate 3000 nanoRSLC UHPLC system (Thermo Scientific, Germeringen, Germany). Samples were injected onto a C18 column (75 µm, 250 mm, 1.6 µm particle size, Aurora, Ionopticks, Fitzroy, Australia) kept at 50 °C. Peptides were loaded at 400 nL/min for 1 min in 3% solvent B and separated by a multi-step gradient: 5% solvent B at 2 min 17%, solvent B at 24 min, 25% solvent B at 29 min, 34% solvent B at 32 min, 99% solvent B at 33 min held until 40 min, returning to 3% solvent B at 40.1 min and held until 58 min to recondition the column (Solvent A: 0.1% formic acid in water, Solvent B: 0.1% formic acid in acetonitrile). MS analysis of eluting peptides was performed by a time-of-flight mass spectrometer. The precursor scan ranged from 100 to 1700 *m*/*z* and a time range of 0.6–1.6 V.s/cm^2^ in PASEF mode. A total of 10 PASEF MS/MS scans following collision induced dissociation (collision energy from 20–59 eV) were collected with a total cycle time of 1.16 s.

Raw MS/MS data were processed using Maxquant software, Dr. Jürgen Cox, Max Planck Institute of Biochemistry, Martinsried, Germany (version 1.6.14.0) [34] searching a pan proteome database of *L. crispatus* (Uniprot, downloaded 1 November 2021). To estimate the false spectrum assignment rate, a reverse version of the same database was also searched. The settings were as follows: Enzyme Trypsin/P allowing for a maximum of two missed cleavages, variable modifications: Oxidation (M), fixed modifications: Carbamidomethyl (C). Settings were default for timsDDA, and match between runs was enabled with a matching time window of 0.2 min and a matching ion mobility window of 0.05 indices. For label free quantification, both iBAQ and LFQ were enabled [34].

### 4.8. Analysis of the PulA Gene Region

The contig of *L. crispatus* RL10 was analyzed with SnapGene version 5.3.2. (GSL Biotech LLC, Chicago, IL, USA). The promoter sequences (−35 and −10 elements) were predicted by the use of BPROM (Prediction of Bacterial Promoters) and FindTerm (Finding Terminators in bacterial genomes) [35]. The catabolite responsive element (*cre* site) was identified by matching the palindromic nucleotide motif of *L. plantarum* (TGWAANCGNTNWCA) [36].

### 4.9. Analysis of Publicly Available Vaginal Metagenomes

The raw shotgun metagenome sequencing data of 1507 publicly available vaginal microbiome samples were downloaded from the VIRGO database [28]. In addition to the 211 samples which were sequenced for the construction of VIRGO, this large data collection contains samples from other studies of North American [37], African [5], and Chinese women [38]. The Illumina reads were processed using FastQC to determine data statistics, such as data yield per sample. The samples were taxonomically characterized using Kraken 2 [39] and Bracken [40]. The human read counts in the species-level results from Bracken were removed and the data were normalized to obtain relative abundances of non-human taxa, such as bacteria, archaea, viruses, and fungi. Taxa that had less than 0.5% relative abundance and which were observed in less than 5% of the samples were filtered out. The downstream taxonomic analyses and visualization were performed using R v3.6.3 (R Core Team, 2020, Robert Gentleman and Ross Ihaka, the ‘R & R’ of the Statistics Department of the University of Auckland, New Zealand with R packages vegan, phyloseq [41], ComplexHeatmap [42], and ggplot2 v3.3.3.

### 4.10. Amylopullulanase Sequence Variant Detection

The raw reads were aligned to the RL10 *pulA* gene sequence using bowtie2 [43] with default parameters except for “--very-sensitive-local --no-unal”. Alignment statistics for each sample, such as coverage were calculated using samtools (John Marshall and Petr Danecek et al Initial release: 2009) v1.13 [44]. Single-nucleotide polymorphisms (SNPs) and short indels in samples with a minimum 10× coverage were determined using FreeBayes v1.2 [45] in a multi-sample variant calling approach using default parameters except for “-min-alternate-count 2 -min-coverage 10 -min-alternate-fraction 0.01 -use-duplicate-reads -min-mapping-quality 0 -ploidy 1 -pooled-continuous”. The variants were annotated in terms of their impact on the RL10 *pulA* protein sequence using SnpEff v4.3t [46]: high (e.g., mutations that result in frameshifts or stop gain), moderate (e.g., mutations that result in an amino acid change), and low (synonymous mutations). The variant allele frequency heatmap for variants that had at least 0.05 alternate allele frequency (i.e., supported by at least 5% of the reads at that position) and were observed in at least three samples was plotted using R packages VariantAnnotation [47], and ggplot2. SNPs and short indels with a minimum frequency of 0.7 were classified as high-frequency short variants.

The same sample set used in small variant detection which had a minimum coverage of 10 times along the RL10 *pulA* sequence was used for detecting larger and structural variations, such as insertions, deletions, inversions, translocations, and duplications in read mappings. To this end, the indels and Structural Variants tool in CLC Genomics v.12.0.3 (Health Sciences Library System, Pittsburgh, PA, USA) was used with default parameters (*p*-value threshold = 0.0001) except for “minimum number of reads = 10”. Structural variants with a minimum frequency of 0.7 were classified as high-frequency large variants.

### 4.11. Identification of PulA Genes with Transposase Insertions

To create a database of *L. crispatus* transposase gene sequences, we first downloaded 147 annotated *L. crispatus* genomes from NCBI (National Center for Biotechnology Information, Bethesda, MD, USA) (assembly level: scaffold, chromosome, and complete). We then created a FASTA file of 19.551 redundant transposase gene sequences by combining the coding sequences which had a ‘transposase’ annotation in the 147 *L. crispatus* genomes. This database was clustered at 95% sequence identity using CD-HIT using default parameters [48] to obtain a non-redundant transposase database of 508 gene sequences.

To identify *pulA*^−^ rich samples that had transposase insertions, we mapped the reads which were mapped to the *L. crispatus* RL10 *pulA* sequence to the above-mentioned transposase database using bowite2 as described above. The list of samples where more than 20 transposase reads were found was compared to the list of samples with high-frequency structural variants to determine the impact of transposase insertions in causing disruptive structural variations.

## Figures and Tables

**Figure 1 ijms-23-05590-f001:**
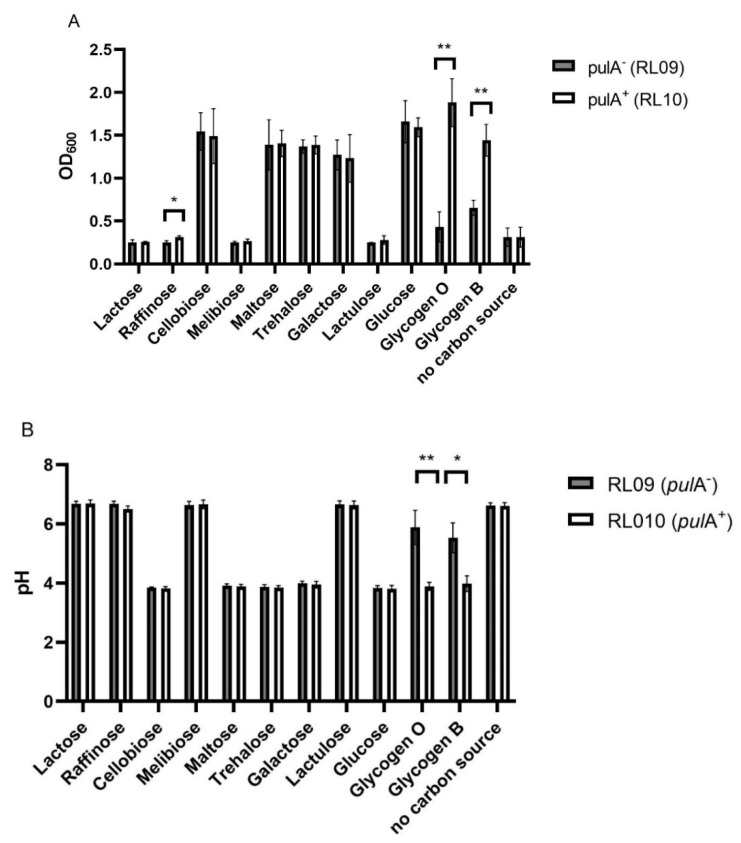
Carbohydrate utilization by *L. crispatus* strains *pulA*^−^ (RL09) and *pulA^+^* (RL10). (**A**) Optical density (600 nm), and (**B**) pH after serial propagation in NYCIII medium supplemented with different carbohydrates. The bars represent the mean of three biological replicates, the standard deviations are shown as error bars. * *p* < 0.05, ** *p* < 0.005 (multiple *t*-test with false discovery rate).

**Figure 2 ijms-23-05590-f002:**
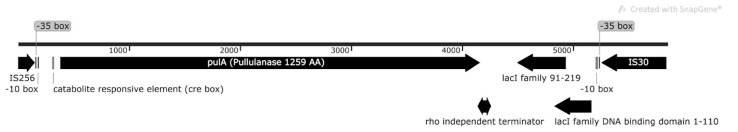
The *pulA* locus (4842 bp) in the chromosome of *L. crispatus* RL10.

**Figure 3 ijms-23-05590-f003:**
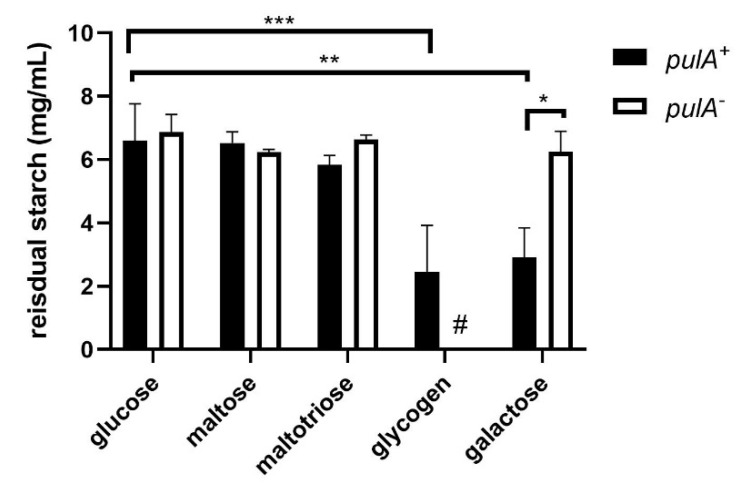
Cell-bound starch degrading activity of *L.*
*crispatus pulA* sufficient and deficient strains grown on NYC-, glucose, maltose, maltotriose, glycogen (only *pulA*^+^) and galactose. Cells were cultured in NYCIII medium supplemented with these carbohydrates and after centrifugation and resuspension of the pellet incubated in a starch solution (7.5 g/L). Asterisks indicate *p*-values calculated by Mann–Whitney of * *p* < 0.05, ** *p* < 0.005, *** *p* < 0.0005. Hashtag (#) indicates no data due to the absence of growth of this strain on glycogen.

**Figure 4 ijms-23-05590-f004:**
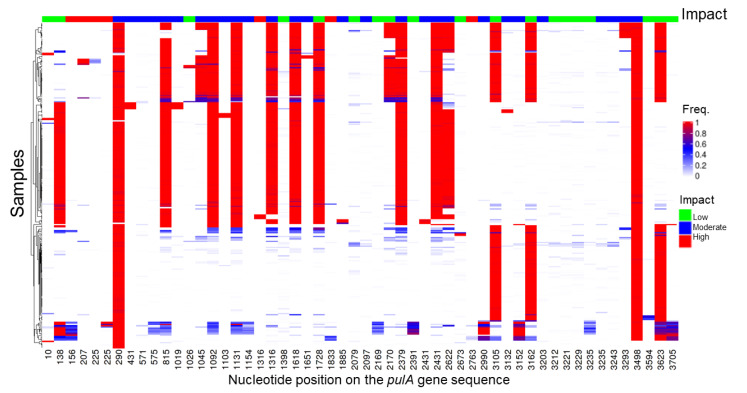
Small variants in the *L. crispatus* amylopullulanase gene in the metagenome of vaginal microbial communities. The list and frequency of SNPs and short indels (small variants) identified in the 270 vaginal microbiome samples that showed at least ten times full length coverage for the pulA gene sequence. The variants were classified in terms of their impact on the protein sequence.

**Table 1 ijms-23-05590-t001:** Levels of SLAP domain containing proteins in the S-layer enriched fraction of *L. crispatus*. Columns in the table include ACC NO, UniProt Accession Number; PEP, number of detected peptides; RL09 GAL, expression levels (iBAQ) in *L. crispatus pulA*^−^ strain RL09 cultivated on galactose as only carbon source, RL10 GAL, expression levels in *pulA*^+^ strain RL09 cultivated on galactose; RL10 GLU, expression levels in *pulA*^+^ strain RL10 cultivated on glucose; RL10 GLY, expression levels in *pulA*^+^ strain RL10 cultivated on glycogen; ^2^LogR_1_, ^2^log ratio of expression levels of strains RL10 and RL09 both cultivated on galactose (fold change of expression calculated from LFQ-values from maxquant); ^2^logR_2_, ^2^log ratio of expression levels of *pulA*^+^ strain RL10 cultivated on glycogen and glucose; ND, not detected (below the detection limit).

ACC NO	Description	PEP	RL09 GAL	RL10 GAL	RL10 GLU	RL10 GLY	^2^logR_1_	^2^logR_2_
A0A4Q0LMV5	Amylopullulanase	64	2.8 × 10^5^	2.5 × 10^7^	7.5 × 10^5^	1.8 × 10^7^	6.5	4.6
A0A135Z5X6	Cell surface protein	11	1.5 × 10^6^	3.2 ×10^6^	1.2 × 10^6^	3.2 × 10^6^	1.1	1.4
D5GXS8	Lysin (Glycoside Hydrolase Family)	10	2.9 × 10^5^	3.6 × 10^5^	4.0 × 10^5^	9.7 × 10^5^	0.3	1.3
D5H1Q3	Aggregation promoting factor	4	ND	ND	1.5 × 10^5^	2.2 × 10^5^	ND	0.6
A0A135ZFI7	N-acetylmuramidase	17	1.3 × 10^6^	2.0 × 10^6^	4.0 × 10^6^	4.1 × 10^6^	0.6	0.04
K1MET7	Uncharacterized protein	29	6.4 × 10^6^	1.0 × 10^7^	7.6 × 10^6^	2.3 × 10^6^	0.7	0.04
A0A6M1GK10	S-layer protein (Fragment)	32	4.2 × 10^5^	6.9 × 10^5^	3.0 × 10^5^	3.0 × 10^5^	0.7	−0.02
C7XI68	SlpA domain-containing protein	14	3.2 × 10^5^	7.3 × 10^5^	6.4 × 10^5^	6.1 × 10^5^	1.2	−0.07
A0A6B8SPR2	SlpA domain-containing protein	15	3.1 × 10^7^	2.2 × 10^7^	1.3 × 10^7^	1.1 × 10^7^	−0.5	−0.2
A0A135ZF64	Amidase	10	5.7 × 10^5^	1.7 × 10^6^	1.7 × 10^6^	1.4 × 10^6^	1.6	−0.2
A0A135ZH73	Lysin	11	4.5 × 10^6^	8.8 × 10^6^	6.6 × 10^6^	5.6 × 10^6^	1.0	−0.2
A0A7V8KTK5	Putative surface layer protein	48	1.6 × 10^7^	2.8 × 10^6^	2.7 × 10^7^	2.2 × 10^7^	0.8	−0.3
E3R4X6	Uncharacterized protein	9	2.6 × 10^6^	2.8 × 10^6^	2.7 × 10^6^	2.3 × 10^6^	0.1	−0.3
K1MJV3	Uncharacterized protein	22	1.6 × 10^7^	2.9 × 10^7^	2.0 × 10^7^	1.6 × 10^7^	0.8	−0.3
V5EI62	Lysin (Glycoside Hydrolase Family)	15	2.4 × 10^6^	4.6 × 10^6^	1.1 × 10^7^	7.6 × 10^6^	0.9	−0.5
V5EF53	SlpA domain-containing protein	12	2.1 × 10^7^	3.0 × 10^7^	6.1 × 10^7^	4.3 × 10^7^	0.5	−0.5
A0A4Q0LUE2	Uncharacterized protein	24	2.7 × 10^7^	1.8 × 10^7^	3.3 × 10^7^	2.3 × 10^7^	−0.6	−0.5
A0A4R6CSS1	Cell surface protein (SlpA)	21	2.4 × 10^6^	5.8 × 10^6^	6.6 × 10^6^	4.6 × 10^6^	1.3	−0.5
D0DJ09	Bacterial surface layer protein	45	3.6 × 10^8^	4.8 × 10^8^	5.1 × 10^8^	3.5 × 10^8^	0.4	−0.5
K1M186	Uncharacterized protein	12	8.0 × 10^6^	9.7 × 10^6^	9.5 × 10^6^	6.1 × 10^6^	0.3	−0.6
A0A135YZ62	Bacterial surface layer protein (SlpA)	32	2.0 × 10^7^	3.3 × 10^7^	8.7 × 10^7^	5.5 × 10^7^	0.8	−0.6
D0DIV9	Uncharacterized protein	36	5.3 × 10^7^	7.6 × 10^7^	1.3 × 10^8^	7.9 × 10^7^	0.5	−0.7
V5GA31	Lactocepin s-layer protein	10	1.8 × 10^6^	2.7 × 10^6^	8.4 × 10^6^	3.2 × 10^6^	0.6	−1.4
A0A120DHV3	Uncharacterized protein	16	6.0 × 10^6^	1.1 × 10^7^	2.1 × 10^7^	7.7 × 10^6^	0.8	−1.4
K1N369	Uncharacterized protein	8	9.8 × 10^4^	3.7 × 10^4^	1.4 × 10^5^	3.0 × 10^4^	−1.4	−2.2
A0A125P6L8	S-layer protein (SlpA)	13	1.1 × 10^4^	1.3 × 10^5^	8.7 × 10^5^	4.2 × 10^3^	3.5	−7.7
A0A7T1TNA6	Bacterial surface layer protein	30	4.7 × 10^6^	5.7 × 10^6^	6.1 × 10^7^	2.5 × 10^5^	0.3	−7.9
A0A1C2D5C0	Cell surface protein	5	ND	1.5 × 10^6^	8.1 × 10^5^	ND	ND	ND
Fraction of total detected proteins (iBAQ)			0.47	0.46	0.22	0.40		

**Table 2 ijms-23-05590-t002:** Overview of amylopullulanase variants in our strain collection of human vaginal *L. crispatus* isolates [13].

Strain	*PulA* Variant
RL03, RL08, RL10, RL11, RL13, RL14, RL15, RL16, RL17, RL20, RL21, RL22, RL23, RL24, RL25, RL27, RL28, RL29, RL30, RL33	Wild type
RL02 and RL09	Two nucleotide deletion
RL06 and RL07	N-terminal insertion of mobile element (variant 1)
RL19 and RL26	N-terminal insertion of mobile element (variant 2)
RL05	Large deletion

## Data Availability

All data presented here are included in the manuscript, Appendix B or Appendix A. The mass-spectrometry data have been deposited in Proteome Exchange.

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
