# Peer review of "Genetic Elements Orchestrating Lactobacillus crispatus Glycogen Metabolism in the Vagina"

_ijms, 2022, doi:10.3390/ijms23105590_

Round 1

Reviewer 1 Report

The manuscript describes an analysis carried out to investigate genetic variation and differential regulation of the pulA gene. Molecular and proteomic methodologies were employed and the studies concluded that the alpha-glucan degradation activity dissipates when Lactobacilllus crispatus is cultured in glucose, maltose and maltotriose, due to catabolite repression that occurs by flanking the pulA gene. This type of metabolic work is important in several areas of knowledge. The manuscript is well presented, with a clear objective, numerically adequate figures and references. The manuscript deserves publication, however is necessary to clarify some points that are confused, before being recommended for publication.

Major points

-Lines 24-26 (Abstract) - “Analysis of the largest 24 publicly available human vaginal metagenome dataset indicates …..”- Don't you think that the percentage (5-10%) of the bacterial population is significantly low, to corroborate your metabolomic and microenvironment deductions?

- Line 61-62 (Introduction)- “The results showing substantial disruption and down-regulation of the Lactobacillus crispatus pullulanase gene.” – Please, explain which experiments allow you to conclude this statement.

- Lines 284-286 (Discussion) –“Our study implies that L. crispatus grown in such an environment will first grow on glucose, maltose and maltotriose and will not express pullulanase until these  carbohydrates are depleted. This will diminish the advantage of cheating or any other evolutionary pressure on the production of alpha-glucosidase” - Please explain the sentence described. I was not able to identify in the studies presented in the paper the set of experiments that actually demonstrate this statement.

Author Response

Major points reviewer 1

-Lines 24-26 (Abstract) - “Analysis of the largest 24 publicly available human vaginal metagenome dataset indicates …..”- Don't you think that the percentage (5-10%) of the bacterial population is significantly low, to corroborate your metabolomic and microenvironment deductions?

> Thank you for your comment. The percentage is not referring to the bacterial population, but to the number of samples with a disrupted pulA gene. In these samples exclusively a disruptive pulA gene has been identified. We have clarified this point by correcting this number in the abstract and addition of Appendix table A3

Abstract

‘Analysis of the largest publicly available dataset of 1507 human vaginal metagenomes indicates that among the 270 samples that contain a Lactobacillus crispatus pulA gene, 62 samples (23%) had a defective variant of this gene’

Table A3

Ethnic group

#samples

#samples with

pulA

#samples with

defective pulA

Total

1507

270

62

Black or African American

781

102

12

White or Caucasian

163

55

19

Chinese

76

4

2

Unknown

472

108

28

Hispanic, mixed or other

15

1

1

Table A3. Number of metagenomes (total 1507), with a pulA gene (intact and defective) and a defective pulA gene per ethnic group of women.

- Line 61-62 (Introduction)- “The results showing substantial disruption and down-regulation of the Lactobacillus crispatus pullulanase gene.” – Please, explain which experiments allow you to conclude this statement.

>Thank you for your comment. We have rephrased this paragraph as below

 ‘Here we present results showing suppression of PulA activity in glucose-, maltose- and maltotriose-grown cultures by enzymatic assays for alpha-glucosidase. We use a mass spectrometric approach to demonstrate the presence of the L. crispatus amylopullulanase enzyme in an S-layer enriched fraction, and its loss in a strain with a defective pulA gene. Analysis of a vaginal metagenomic dataset demonstrates the presence of a significant number of mutations and structural variants expected to fully disrupt amylopullulanase functionality in vivo, indicating that both environmental as well as genetic factors may refrain vaginal L. crispatus from metabolizing glycogen.’  

- Lines 284-286 (Discussion) –“Our study implies that L. crispatus grown in such an environment will first grow on glucose, maltose and maltotriose and will not express pullulanase until these  carbohydrates are depleted. This will diminish the advantage of cheating or any other evolutionary pressure on the production of alpha-glucosidase” - Please explain the sentence described. I was not able to identify in the studies presented in the paper the set of experiments that actually demonstrate this statement.

 > We have clarified this point as follows.

‘The results of our alpha-glucosidase assays and proteome analysis imply that L. crispatus grown in such an environment will first grow on glucose, maltose and maltotriose and will suppress amylopullulanase expression until these carbohydrates are depleted. This will diminish the advantage of cheating or any other evolutionary pressure on the production of alpha-glucosidase.’

Reviewer 2 Report

In my opinion, the scope of the presented research is interesting and the research itself is well methodical.The manuscript looks really good in terms of content and graphics.I think the work will be interesting for IJMS readers.

However, I have a few minor suggestions for the official approval of the manuscript for print:

1) Table 1 - please give an explanation to ND

2) Figs. 4, A3 - please improve quality / readability

Overall a good job!

Author Response

Reviewer 2

In my opinion, the scope of the presented research is interesting and the research itself is well methodical.The manuscript looks really good in terms of content and graphics.I think the work will be interesting for IJMS readers.

However, I have a few minor suggestions for the official approval of the manuscript for print:

1)    Table 1 - please give an explanation to ND

> Thank you for comment. We have added ND, not detected (below the detection limit). to the table legend.

2)    Figs. 4, A3 - please improve quality / readability

> Thank you for your comment. Figure A3 is not very informative and indeed, not easy to read, thus we have decided to remove this figure from the revised version of the manuscript and replace this by a Appendix table A3 (see response to reviewer 1) 

 Overall a good job!

> Thank you